# The Sigma Receptors in Alzheimer’s Disease: New Potential Targets for Diagnosis and Therapy

**DOI:** 10.3390/ijms241512025

**Published:** 2023-07-27

**Authors:** Tao Wang, Hongmei Jia

**Affiliations:** 1Key Laboratory of Radiopharmaceuticals (Beijing Normal University), Ministry of Education, College of Chemistry, Beijing Normal University, Beijing 100875, China; 201831150053@mail.bnu.edu.cn; 2Department of Nuclear Medicine, Xinqiao Hospital, Army Medical University, Chongqing 400037, China

**Keywords:** sigma-1 receptor, sigma-2 receptor, Alzheimer’s disease, positron emission tomography, neuroimaging, diagnosis, therapeutic strategy

## Abstract

Sigma (σ) receptors are a class of unique proteins with two subtypes: the sigma-1 (σ_1_) receptor which is situated at the mitochondria-associated endoplasmic reticulum (ER) membrane (MAM), and the sigma-2 (σ_2_) receptor, located in the ER-resident membrane. Increasing evidence indicates the involvement of both σ_1_ and σ_2_ receptors in the pathogenesis of Alzheimer’s disease (AD), and thus these receptors represent two potentially effective biomarkers for emerging AD therapies. The availability of optimal radioligands for positron emission tomography (PET) neuroimaging of the σ_1_ and σ_2_ receptors in humans will provide tools to monitor AD progression and treatment outcomes. In this review, we first summarize the significance of both receptors in the pathophysiology of AD and highlight AD therapeutic strategies related to the σ_1_ and σ_2_ receptors. We then survey the potential PET radioligands, with an emphasis on the requirements of optimal radioligands for imaging the σ_1_ or σ_2_ receptors in humans. Finally, we discuss current challenges in the development of PET radioligands for the σ_1_ or σ_2_ receptors, and the opportunities for neuroimaging to elucidate the σ_1_ and σ_2_ receptors as novel biomarkers for early AD diagnosis, and for monitoring of disease progression and AD drug efficacy.

## 1. Introduction

Alzheimer’s Disease (AD) is the most common form of dementia, accounting for 60–80% of all cases. According to the World Alzheimer Report 2022, about 55 million people lived with dementia worldwide in 2019. This number is expected to rise to 139 million in 2050 [1]. AD is characterized by two pathological hall markers—extracellular β-amyloid (Aβ) plaques and intraneuronal fibrillary tangles (NFTs) composed of hyperphosphorylated tau proteins. Based on the 2018 National Institute on Aging—Alzheimer’s Association (NIA-AA) research framework, biomarkers of AD are grouped into those for Aβ (A), pathologic tau (T), and neurodegeneration or neuronal injury (N) [2]. Currently, the ATN research framework, grounded on the above biomarker-based definition of AD, has been widely accepted for diagnosis of AD in clinical practice worldwide [2,3]. However, the exact cause of AD has not been fully elucidated, although many hypotheses on its etiology and pathogenesis have been put forward, including the Aβ cascade hypothesis [4], the misfolded tau protein hypothesis [5], the cholinergic hypothesis [6,7,8,9], oxidative stress [10], calcium dyshomeostasis [11,12], neuroinflammation [13,14] and the mitochondria cascade (or MAM) hypothesis [15,16].

Medications currently available on the market for AD treatment are listed in Table 1. Donepezil is a piperidine-based reversible inhibitor of acetylcholinesterase (AChE) with high inhibitory activity (*IC*_50_(AChE) = 5.7 nM) and high selectivity over butyrylcholinesterase (BuChE) (*IC*_50_(BuChE) = 7138 nM, 1252-fold) [17]. Interestingly, donepezil has also been shown to bind to the sigma-1 (σ_1_) receptors in the living human brain at therapeutic doses [18]. Rivastigmine is a carbamate-based pseudo-irreversible (slowly reversible) dual inhibitor of both AChE and BuChE with low inhibitory potency (*IC*_50_(AChE) = 32,100 nM, *IC*_50_(BuChE) = 390 nM, 82-fold) [19]. However, this drug has no affinity for muscarinic, α- or β-adrenergic, or dopamine (DA) receptors or opioid binding sites [20]. Galantamine is a reversible competitive inhibitor for AChE (*IC*_50_(AChE) = 350 nM) rather than BuChE (*IC*_50_(BuChE) = 18,600 nM, 53-fold) [21], and an allosteric modulator of nicotinic acetylcholine receptors [22]. The most common adverse events of these cholinesterase inhibitors are gastrointestinal (GI) and cardiovascular side effects. They are generally well-tolerated and all are still considered for first-line, symptomatic treatment of AD (for review, see [23,24,25,26,27]).

Memantine is a voltage-dependent, low affinity/fast off-rate and non-competitive *N*-methyl-*D*-aspartate receptor (NMDA) receptor antagonist [28]. It exerts its neuronal protective effects by inhibiting glutamate activity and is used for the treatment of moderate-to-severe AD alone or in combination with donepezil [29,30,31]. Adverse effects of memantine have been found to be comparable to those with a placebo, with the exception of an increased incidence of dizziness, headache, confusion, and constipation [32].

Both Aducanumab (Aduhelm) and Lecanemab (Leqembi) are anti-amyloid monoclonal antibodies (mAbs) and approved under the accelerated approval pathway for treatment of Alzheimer patients with mild cognitive impairment (MCI). On 6 July 2023, the US Food and Drug Administration (FDA) converted Leqembi to traditional approval. And thus, Leqembi represents the first Aβ-directed antibody fully approved for the treatment of AD without any restrictions. Aducanumab is a human IgG1 monoclonal antibody preferably targeting Aβ aggregates [33]. Lecanemab is a humanized monoclonal IgG1 of the mouse mAb158 selectively binding to soluble Aβ protofibrils [34]. Adverse effects from both medications include amyloid-related imaging abnormalities (ARIA) and infusion reactions [33,34,35,36]. Patients with apolipoprotein E ε4 (ApoE ε4) gene carriers, especially ApoE ε4 homozygotes, are at higher risk of ARIA such as brain swelling (edema or effusions) and bleeding (hemosiderin deposits) [36,37,38].

Sodium Oligomannate Capsules (GV-971) were approved by the National Medical Products Administration (NMPA) of China in 2019 for treatment of mild-to-moderate AD [39]. GV-971 was noted to cause an induced liver injury side effect [40]. The other side effects of GV-971 have not been extensively reported in the literature. However, international multicenter clinical trials of GV-971 (Phase III) were stopped in May 2022.

Recent evidence has pointed to the significance of sigma receptors in AD. The σ_1_ receptor, situated at the mitochondria-associated endoplasmic reticulum (ER) membrane (MAM), directly regulates Aβ generation in the MAM [41]. The sigma-2 (σ_2_) receptor has been positively identified as the ER-resident transmembrane protein 97 (TMEM97) [42], and the σ_2_ receptor/TMEM97, progesterone receptor membrane component 1 (PGRMC1) and low-density lipoprotein receptor (LDLR) have been found to form a trimeric complex and regulate the uptake of lipoproteins such as LDL and apolipoprotein E (ApoE) [43,44], whose E4 allele (ApoE ε4) is the greatest risk factor for AD development [45]. Further, this trimeric complex has been demonstrated to be a binding site for Aβ oligomers (AβOs). Inhibition of one of the three proteins results in disruption of the complex and decreased AβO uptake in neurons [44]. As a result, both the σ_1_ and σ_2_ receptors are increasingly viewed as playing critical roles in AD pathogenesis and progression, and are thus important targets for therapeutic intervention to inhibit Aβ neurotoxicity, neurodegeneration, and progression of AD [46,47,48,49,50,51,52,53].

## 2. Sigma Receptors in AD

### 2.1. The Sigma (σ) Receptors

The sigma (σ) receptors were initially identified in 1976 and thought to belong to the class of “opioid receptors” [54]. They were later found to possess binding sites distinct from those of opioid receptors [55], and divided into two subtypes termed σ_1_ and σ_2_ [56], based on the different binding sites of the radioligands (+)-[^3^H]pentazocine and [^3^H]1,3-di(2-tolyl) guanidine ([^3^H]DTG) [57]. Both σ_1_ and σ_2_ receptors are widely distributed in the central nervous system (CNS) [57,58] and peripheral tissues [59,60,61,62], acting as integral membrane proteins and playing crucial roles in a variety of human diseases [63]. However, the regional expression patterns of σ_1_ and σ_2_ receptors in the brain are clearly different [57,64,65,66]. Recent quantitative autoradiography studies with postmortem human brain tissues found higher concentrations of σ_2_ than σ_1_ receptor in all brain regions examined, except the red nucleus, as well as upregulation of σ_2_ receptors in the aged brains [67,68].

The σ_1_ receptor, consisting of 223 amino acids with molecular weight of 25.3 kDa [69], has been cloned from several tissues, including those from mice, rats and guinea pigs [70,71,72,73], and proved to be a unique “ligand-operated receptor chaperone” that is regulated by the agonist/antagonist activity of endogenous or synthetic ligands [74,75]. The crystal structure of the human σ_1_ receptor was elucidated in 2016, and found to have a trimeric structure with a single transmembrane domain in each protomer and a β-barrel cupin fold in the carboxy terminal domain [76]. Currently, there is no consensus on endogenous ligands for the σ_1_ receptor (for review, see [63]), even though some candidates such as the hallucinogen *N*, *N*-Dimethyltryptamine [77] have been proposed.

Compared with the σ_1_ receptor, the identification process for the σ_2_ receptor is more convoluted. In 1994, Bowen et al. employed [^3^H]azido-DTG to estimate the molecular weight of σ_2_ receptor isolated from rat liver membrane, at 21.5 kDa [60]. Subsequently, this enigmatic protein has been hypothesized to contain a histone binding site [78,79]. The σ_2_ receptor complex was also found to contain PGRMC-1 protein complex [80,81]. Then, in 2017, the σ_2_ receptor was positively identified as the four-domain TMEM97 (also known as meningioma-related protein 30, MAC30), residing in the ER membrane [42]. Finally, in 2021, Alon et al. successfully determined the crystal structure of the bovine σ_2_ receptor using high-affinity ligands [82]. The σ_2_ receptor/TMEM97 is now revealed as an intimately associated homodimer, with each of the two protomers having four kinked transmembrane helix sections, and both the N and C terminals facing the cytoplasm. The binding site of the σ_2_ receptor ligand is deeply embedded in the membrane, which suggests that a lipid may be the endogenous ligand [82]. The binding pocket opens laterally into the lipid bilayer, and its opening is lined with hydrophobic and aromatic residues [82]. Up to date, two putative endogenous ligands, histatin-1 [83] and 20(*S*)-hydroxycholesterol (20(*S*)-OHC) [84], have been reported.

### 2.2. Sigma-1 Receptor in AD

The σ_1_ receptor participates in various physiological and pathological processes, such as neurotransmission, neuroprotection and neuroinflammation, through interaction with diverse ion channels, ER proteins, neurotrophins and G protein-coupled transporters (GPCRs) [85]. Consequently, the σ_1_ receptor has been considered as a therapeutic target for a range of diseases [50,86] including amnesia and AD [46,85,87,88], Parkinson’s disease (PD) [89,90,91,92,93], Huntington’s disease (HD) [94,95], amyotrophic lateral sclerosis (ALS) [96,97], retinal disease [98,99,100,101], multiple sclerosis (MS) [102], major depressive disorder (MDD) [89,103], stroke [104,105,106], pain [107,108] and drug or alcohol addiction [109]. In cancers, the σ_1_ receptor is involved in tumor occurrence, development, metastasis and survival [110,111,112,113].

Increasing evidence has proved that the σ_1_ receptor holds great potential as a biomarker for early AD diagnosis and progression and the monitoring of AD drug efficacy [49]. Hallmarks of human AD include progressive cognitive decline that follows chronic neuroinflammation and the emergence of hyperphosphorylated tau protein aggregates and Aβ plaques [46], with all playing critical and perhaps interrelated roles in the progression of AD. In particular, Ca^2+^ plays a critical role in learning and memory processes [114,115,116,117,118], and Ca^2+^ dyshomeostasis is a pathological feature of AD and other neurodegenerative diseases [114,115,116,117,118]. The σ_1_ receptor forms a Ca^2+^-sensitive chaperone complex with the binding immunoglobulin protein/glucose-regulated protein 78 (BiP/GRP78), and prolongs Ca^2+^-signaling from ER into mitochondria by stabilizing inositol 1,4,5-trisphosphate receptor (IP_3_R) at the MAM [74]. The σ_1_ receptor participates in the regulation of intracellular Ca^2+^ migration and maintains homeostasis and protects cognitive function damage [119,120]. Research on the role of the σ_1_ receptor in mediating mitochondrial function has found that the σ_1_ receptor attenuates hippocampal dendrite formation through scavenging of free radicals, and protects cells from damage by mitochondria-derived reactive oxygen species (ROS) [121,122,123]. The ER stress sensor inositol-requiring enzyme 1 (IRE1) facilitates mitochondrion-ER-nucleus signaling for cellular survival via the σ_1_ receptor chaperone [75,124].

More importantly, the σ_1_ receptors regulate early Aβ generation in AD at the MAM [41], and thus could be considered as a bona fide MAM marker and responsible for neuroprotective regulatory functions [87]. It has been proposed that σ_1_ receptor comprises part of the endogenous cellular defense against toxic Aβ [85,87]. Growing evidence has demonstrated the neuroprotective activity of the σ_1_ receptor against Aβ neurotoxicity (for review, see [46,125]). Activation of σ_1_ receptor potentiates nerve growth factor (NGF)-induced neurite outgrowth through modulating the PLCγ-DAG-PKC, Ras-Raf-MEK-ERK-MAPK signaling pathways and protects Aβ_25–35_-impaired dentritic growth and survival of newborn neurons through a modulation of PI3K-Akt-mTOR-p70S6k signaling [119,126,127]. Moreover, activating the σ_1_ receptor increases vascular endothelial growth factor (VEGF) and low-density lipoprotein receptor-related protein 1 (LRP-1) expression levels and attenuates the blood–brain barrier (BBB) dysfunction caused by amyloid deposition in AD [48].

Additionally, the mixed muscarinic/σ_1_ ligand ANAVEX2-73 prevents tau hyperphosphorylation in Aβ_25–35_-injected mice [128]. Further, the σ_1_ receptor regulates proper tau phosphorylation and axon development by promoting p35 turnover via σ_1_ receptor–myristic acid interaction, thereby avoiding cyclin-dependent kinase 5 (CDK5)/P25 overactivity [129,130].

Studies with σ_1_ receptor knock-out mice showed that σ_1_ receptor deletion resulted in neurogenesis impairment [131] and cognitive dysfunction [132] including memory deficit, neurocyte susceptibility to Aβ-mediated toxicity and impairment to the intracellular lipid metabolism and immune response, and thus accelerated neural degeneration and oxidative stress-induced neural death [132]. Emerging evidence indicates that certain polymorphisms of the σ_1_ receptor gene, especially when present alongside the known AD risk factor ApoE ε4, are linked to the onset of AD neurodegeneration [133]. In a postmortem study, reduction in σ_1_ receptors was observed in the hippocampus of patients with AD [134]. Compounds with σ_1_ agonist activity have been shown to possess anti-amnestic and neuroprotective efficacy in both pharmacological and pathological AD models [119], including those resulting from cholinergic destruction [135,136], Aβ administration [135,137,138,139,140,141], glutamatergic/serotonergic [142] or calcium channel deficits [143] and normal aging [144], as well as senescence-accelerated mouse (SAM) model [145].

Taken together, it becomes increasingly evident that the σ_1_ receptor plays a key role in mediating AD pathology, and therefore presents as promising therapeutic target for AD.

### 2.3. Sigma-2 Receptor in AD

Compared to the σ_1_ receptor, there are only a few reports on the cellular and molecular biological roles of the σ_2_ receptor. It functions as a housekeeping protein under normal settings [146]. The σ_2_ receptor/TMEM97, a member of the expanded emopamil binding protein (EPR) superfamily, has sterol isomerase activity [147,148] and plays a critical role in cholesterol biology, with correlated expression genes taking part in lipid metabolism [147]. High expression of the σ_2_ receptor was found in a variety of tumor cells, with nearly 10-fold higher expression in a proliferating state tumor compared to a quiescent state [149,150,151,152]. The differential expression of σ_2_ receptors is associated with tumor stage, metastasis, and survival. As such, the σ_2_ receptor can act as a novel biomarker for tumor proliferation [152], and is thus a candidate target for the diagnosis and treatment of common hyperplastic tumors [153]. Further, results from recent studies have indicated the critical involvement of the σ_2_ receptor in the pathophysiology of many brain disorders. Thus, the σ_2_ receptor has been proposed as a novel therapeutic target for AD, HD, PD [147,154,155], depression [156], schizophrenia [157], neuropathic pain [158,159], and age-related macular degeneration [148].

In AD pathology, the σ_2_ receptor interacts with PGRMC1 and LDLR to block AβO from binding neuronal synapses and regulates cholesterol homeostasis [44], and acts as a novel biomarker for AD diagnose and drug development [160].

Recent consensus regards AβO as one of the most toxic and pathogenic forms of Aβ, and elevated AβO levels in the brain as the key causative factor in the formation of Aβ plaques [161]. Studies have shown that AβO-induced neurotoxicity subsequently caused synaptic injury and hampered synaptic plasticity, resulting in abnormalities in synaptic composition, structure and density [162]. The effects of AβO on receptors and signaling pathways are neurodegenerative changes, neuronal injury, synaptic dysfunction and neurofibrillary tangles (NFTs), which eventually lead to memory, learning and cognitive dysfunction. Compared with Aβ monomers and Aβ fibrils, soluble AβOs are more likely to induce neuronal loss and cognitive deficits in amyloid precursor protein (APP)/tau transgenic mice, and their concentrations correlated better with AD severity [163]. Hence, the prevention or reversal of AβO-induced neurotoxicity is thought to be key to AD treatment.

Several lines of evidence have pointed to the critical involvement of the σ_2_ receptor in mediating AβO neurotoxicity and thus the key role it plays in AD pathogenesis and progression [67]. Synthetic AβOs derived from the brains of AD patients were discovered to attach to nerve cells and display typical receptor–ligand pharmacological interaction [160,164]. AβOs specifically and saturably bound to hippocampal and cortical neurons both in vivo and in vitro. AβO treatment induced progressive upregulation of σ_2_ receptor expression in neurons, with more intense AβO binding associated with higher σ_2_ expression. Selective σ_2_ receptor modulators competitively inhibited/reversed AβO binding to neurons, and prevented synapse loss in a dose-dependent manner both in vitro, and in rat models of AD [164].

The σ_2_ receptor regulates the binding and signal transmission of AβO in CNS, and its antagonists can decrease AβO binding to nerve cells and disassociate the attached AβO from neurons [53,160]. Studies have found that the σ_2_ receptor/TMEM97, PGRMC1 and LDLR can form a ternary complex (σ_2_R-PGRMC1-LDLR) [43], which is a binding site for monomeric and oligomeric amyloid Aβ_42_, and plays an essential role in the uptake of fibers and oligomers via ApoE-dependent and independent mechanisms [44]. The knockout of the TMEM97/σ_2_ receptor, or PGRMC-1, or both, as well as inhibition of the TMEM97/σ_2_ receptor were all shown to reduce the uptake of Aβ_1–42_ and ApoE in primary neurons [44]. Moreover, the expression of the σ_2_ receptor is dramatically increased by approximately 1.5-fold in AD [165], and is localized to an increased area of synapses (approximately 1.8-fold) in brain tissue taken from people suffering from AD compared with healthy controls, suggesting a compensatory response to AD-related synaptic depression [148,165]. Neurons with knockout of the PGRMC-1 protein also displayed reduced capacity in binding to AβO [160]. The σ_2_ receptor/TMEM97 is present in synaptic fractions biochemically isolated from human temporal cortex, and its concentrations appeared to be higher in samples isolated from AD patient brains compared to those from healthy controls [165].

As a cholesterol-regulating protein [51], the malfunctions of the σ_2_ receptor/TMEM97 are involved in AD pathology [51,52]. The σ_2_ receptor ligands also potentially influence Aβ synthesis via cholesterol, which has been demonstrated to directly affect APP cleavage in neuronal cultures by boosting β- and γ-secretase activity [166]. In CNS, the neurons obtain cholesterol mostly via multiple ApoE receptors including LDLR, very-low-density lipoprotein receptor (VLDLR), and LDLR-related protein 1 (LRP1) [167]. High cholesterol levels are recognized to be a risk factor for AD [168]. The σ_2_ receptor ligands can potentially interrupt lipoprotein transport [167], decrease the level of cholesterol and exert anti-AD effects. Similarly, σ_2_ receptor ligands can influence tau phosphorylation via cholesterol [169]. Simultaneously, the hyperphosphorylated tau is found in lipid rafts, implying that cholesterol has the ability to control tau hyperphosphorylation [170]. More recently, a close physical colocalization of TMEM97 and TSPO was found in MP cells. The σ_2_ receptor ligands such as siramesine modulated TMEM97-TSPO association [171]. In addition, σ_2_ receptor ligands have been also reported to activate liver X receptors (LXRs) through oxysterols and inhibit the expression of inflammatory genes, thereby regulating neuroinflammation in AD [147,172]. Therefore, the σ_2_ receptor is a novel regulator of cholesterol homeostasis in the AD pathological process, and its ligands may target cholesterol homeostasis for AD treatment [147].

Similar to the σ_1_ receptor, the σ_2_ receptor is involved in the regulation of intracellular Ca^2+^ levels [173,174,175]. Binding of AβO to neurons upregulates the σ_2_ receptor in AD and triggers ER stress, to disrupt Ca^2+^ homeostasis. Antagonism of the σ_2_ receptor is believed to reduce ER stress, maintain Ca^2+^ homeostasis and protect neurons [176]. Small molecules acting at the σ_2_ receptor have also been shown to exert their neuroprotective activity via regulation of neuroinflammation and the nerve growth factor (NGF) [176,177,178].

Taken together, the intact σ_2_ receptor/TMEM97-PGRMC1-LDLR complex is a pathway for the cellular uptake of AβO via ApoE-dependent and independent mechanisms. The loss or pharmacological inhibition of one or both of these proteins results in the disruption of the complex leading to decreased uptake of AβO and ApoE in neurons. Targeting the σ_2_ receptor/TMEM97 represents a new strategy for inhibiting Aβ neurotoxicity and slowing neurodegeneration in AD [53,148].

## 3. Ligands Targeting σ_1_ or σ_2_ Receptors

### 3.1. Agonists/Antagonists of σ Receptors and Their Therapeutic Potential in Clinical Trials

Many ligands targeting σ_1_ and σ_2_ receptors have been investigated for their therapeutic potential [46,49,50,179,180,181,182,183,184,185]. Representative agents in clinical trials are provided in Figure 1 and Table 2. The progress of these ligands has been covered in some recent reviews [50,185]. Here, we describe only the ligands with therapeutic potential for AD.

Dextromethorphan (AVP-923), a σ_1_ receptor agonist, has been found to have multiple mechanisms of action that could be beneficial in AD, such as anti-inflammatory and antioxidant effects, modulation of neurotransmitters, and a neuroprotective effect by inhibiting Aβ aggregation and tau hyperphosphorylation in AD [186,187,188]. ANAVEX2-73 (blarcamesine), a σ_1_ receptor agonist, has been shown in Phase II clinical trials to provide significant and sustained improvement in cognitive function and reduce neurodegenerative pathology in mild-AD patients [189]. Endonerpic maleate (T-817MA), an orally available neurotropic drug with high affinity to the σ_1_ receptor [50,190], attenuates Aβ-induced neurotoxicity and memory deficits, promotes neurite outgrowth, and preserves hippocampal synapses, likely via σ_1_ receptor activation [191,192]. CT1812, a σ_2_ receptor allosteric antagonist for mild-to-moderate AD treatment [193], has proved to displace toxic AβO from the synaptic receptor, facilitate oligomer clearance into the CSF and restore cognitive function [193,194].

It is important to note that while these σ receptor ligands have shown promising results in preclinical studies, their efficacy and safety in human clinical trials are still being evaluated. It will require further research and development to determine their potential as therapeutic options for AD.

**Table 2 ijms-24-12025-t002:** Representative σ_1_ and σ_2_ ligands in clinical trials ^a^.

Agents	Property	Disease	Clinical Trials ID	Phase/Status
Pridopidine	σ_1_ receptor agonist	HD	NCT03019289 NCT00724048 NCT04556656 NCT00665223 NCT02006472 NCT01306929 NCT02494778	I/Completed II/III/Completed III/Recruiting III/Completed II/Completed II/Completed II/Terminated
		Levodopa-induced dyskinesia (PD)	NCT03922711	II/Terminated
		ALS	NCT04297683 NCT04615923	II/III/Recruiting II/III/Active, not recruiting
Dextromethorphan (AVP-923) ^b^	σ_1_ receptor agonist mu (μ) opioid agonist and NMDA receptor antagonist [195]	AD	NCT00788047 NCT01584440 NCT01832350 NCT02446132 NCT02442778 NCT02442765 NCT00726726 NCT04947553 NCT05557409 NCT04797715 NCT00056524	I/Completed II/Completed IV/Terminated III/Recruiting III/Completed III/Completed I/Completed III/Recruiting III/Recruiting III/Completed III/Completed
SA4503	σ_1_ receptor agonist	Ischemic stroke	NCT00639249	II/Completed
		MDD	NCT00551109	II/Completed
MR309 (E-52862)	σ_1_ receptor antagonist	Oxaliplatin-induced neuropathy	Ref. [196]	IIa/Completed
ANAVEX2-73 (blarcamesine)	σ_1_ receptor agonist muscarinic receptor modulator	Moderate AD	NCT04314934 NCT03790709 NCT02756858 NCT02244541	IIb/III/Recruiting IIb/III/Completed II/Completed IIa/Completed
		Rett syndrome	NCT04304482 NCT03941444 NCT03758924	II/Recruiting III/Completed II/Completed
		PD	NCT03774459	II/Completed
Edonerpic maleate (T-817MA)	σ_1_ receptor activation	Mild-to-moderate AD	NCT00663936 NCT04191486	II/Completed II/Recruiting
		Aβ inhibitor	NCT02079909	II/Completed
		Hepatic impairment	NCT02693197	I/Completed
Roluperidone (MIN-101)	σ_2_ receptor antagonist and 5-HT_2A_ receptor antagonist	Negative symptoms of schizophrenia	NCT03397134	III/Completed
			NCT02232529	I/Completed
		Schizophrenia	NCT03038646	I/Completed
		Healthy subjects	NCT03072056	I/Completed
CT1812	σ_2_ receptor antagonist	Healthy volunteers	NCT03716427	I/Completed
		AD	NCT05531656	II/Not recruiting
			NCT04735536	II/Completed
			NCT02907567	I/II/Completed
			NCT05248672	I/Completed
			NCT05225389	I/Completed
			NCT03507790	II/Recruiting
			NCT03493282	I/II/Completed
			NCT03522129	I/Completed
		Age-related macular degeneration	NCT05893537	II/Recruiting
		Dementia with Lewy bodies	NCT05225415	II/Recruiting
		Cognitive impairment	NCT02570997	I/Completed

^a^ The clinical trials were obtained from https://www.clinicaltrials.gov (accessed on 14 July 2023). ^b^ There are 115 clinical trials for Dextromethorphan. Only AD-related trials are listed.

### 3.2. Development of Radioligands for Neuroimaging of σ Receptors

#### 3.2.1. Characteristics of Optimal σ_1_ or σ_2_ Receptor Radioligands for PET Imaging in AD

Non-invasive radioligand-based molecular imaging technique such as positron emission tomography (PET) imaging is a powerful tool for the investigation of protein target expression and function in living subjects. It can visualize molecular biological processes in normal and disease states [197]. PET imaging of AD pathologic biomarkers such as Aβ and tau has been widely used for evaluating the pathologic features of AD, tracking AD progression, monitoring therapeutic interventions and facilitating drug development based on the ATN research framework [198]. Increasing evidence in recent years has proved that the σ_1_ and σ_2_ receptors play significantly distinct roles in AD pathology [46,147]. In vivo visualization of the σ_1_ and σ_2_ receptor changes in the progression of AD with PET radioligands will shed new light on the involvement of these receptors in the etiology and pathophysiology of AD, and provide a tool to monitor the treatment effect of σ_1_ and σ_2_ receptor-targeted therapeutic agents.

Similar to other neuroimaging radioligands, the development of suitable PET radioligands targeting the σ_1_ or σ_2_ receptor is a great challenge, due to the limited information of the target protein in the brain and presence of the blood–brain barrier (BBB). The optimal radioligands for imaging of the σ_1_ or σ_2_ receptors in the brain need to meet the following requirements: (1) appropriate affinity for the σ_1_ or σ_2_ receptors and high selectivity over other receptors, transporters and ion channels (> 50-fold); (2) an efficient method for radiosynthesis, with good radiochemical yield and high molar activity; (3) suitable physical–chemical properties, including desirable lipophilicity (log *D* = 1–3) and in vitro stability; (4) high brain uptake (SUV > 1) and high brain-to-blood ratios; (5) excellent in vivo stability without radioactive metabolites able to enter the brain; (6) appropriate pharmacological properties that reflect the regional expression of σ_1_ or σ_2_ receptors in the brain; (7) high specific binding to the σ_1_ or σ_2_ receptors in vivo in brain tissue; (8) suitable kinetic (reversible binding) in the human brain; and (9) acceptable toxicological properties, with no side effects in the range of injectable doses.

#### 3.2.2. Radioligands Targeting the σ_1_ Receptor

Over the last two decades, many efforts have been devoted to the development of PET radioligands for the σ_1_ receptors. However, only a few radioligands have been investigated in non-human primates and humans, due to the difficulties in meeting the critical requirements outlined above [49,153,199,200,201]. They are depicted in Figure 2 and reviewed below.

[^11^C]SA4503 ([^11^C]**1**) was the first PET radioligand used for imaging the σ_1_ receptor in humans [202]. SA4503 was reported as a σ_1_ receptor agonist with high affinity and subtype selectivity over the σ_2_ receptor, and with low affinity for 36 other target proteins in the brain, except for the vesicular acetylcholine transporter (VAChT), with moderate affinity (*K*_i_ = 50.2 nM) [203,204,205]. Later, several groups reinvestigated the binding properties of SA4503 and reported slightly different affinities and subtype selectivity (*K*_i_(σ_1_) = 3.3–4.6 nM; *K*_i_(σ_2_) = 51–242 nM; *K*_i_(σ_2_)/*K*_i_(σ_1_) = 14–55) [203,206,207,208]. Density (*B*_max_) of the σ_1_ receptor was estimated to be 30–600 fmol/mg protein (approximately 3–60 nM) in the human brain [134,209,210]. Radioligands with nanomolar affinity (1–6 nM) appear to be suitable for σ_1_ receptor imaging in the brain, suggesting that SA4503 has a suitable range of affinity for quantitative in vivo imaging. Studies in rodents, cats, monkeys and humans indicated its potential to map σ_1_ receptors in the brain [58,202,211,212,213,214,215]. As a result, [^11^C]SA4503 has been used to investigate the σ_1_ receptor density in the brains of patients with AD [216,217] and PD [218], and σ_1_ receptor occupancy by fluvoxamine [219] and donepezil [18] at clinical doses. It should be noted that two studies with [^11^C]SA4503 to image σ_1_ receptor density in the brains of patients with AD have reported discrepant results. In an initial study, decreased accumulation of [^11^C]SA4503 was observed in the brains of AD patients, seemingly indicating downregulation of the σ_1_ receptor in AD [202,216]. However, a recent study using the same radioligand clearly demonstrated increased σ_1_ receptor expression (approximately 27% increase) in the brains of early-onset AD patients [217]. It was proposed that the lower σ_1_ receptor in AD observed in the previous study [216] may be attributed to the individuals treated with donepezil, which has high affinity for the σ_1_ receptor [220], and thus can act as a blocking agent to decrease the binding of [^11^C]SA4503 [217]. Despite these uses of [^11^C]SA4503 in clinical imaging research, it is not an optimal radioligand for neuroimaging, because of its relatively low subtype selectivity over the σ_2_ receptor (*K*_i_(σ_1_) = 3.3–4.6 nM; *K*_i_(σ_2_) = 51–63 nM; *K*_i_(σ_2_)/*K*_i_(σ_1_) = 14–15) [206,207,208] and relatively slow kinetics for a ^11^C-labeled radioligand in non-human primates and humans. Further, the use of [^11^C]SA4503 needs an on-site cyclotron, due to the short half-life of ^11^C (*T*_1/2_ = 20.4 min). Therefore, more recent efforts have been focused on the search for an optimal radioligand labeled with the longer-lived radionuclide ^18^F (*T*_1/2_ = 109.8 min), and thus are more amenable to translation into clinics for diagnostic applications and multicenter studies of drug-target occupancy and the dose–efficacy relationship.

Among the ^18^F-labelled radioligands shown in Figure 2, [^18^F]FTC-146 ([^18^F]**2**) [221,222] and [^18^F]FPS ([^18^F]**3**) [223] were found to display irreversible kinetics in the human brain and therefore are not suitable for neuroimaging applications.

Radioligands [^18^F]**4**, [^18^F]**5** [224], (*S*)-(–)-[^18^F]fluspidine ([^18^F]**6**) and (*R*)-(+)-[^18^F]fluspidine ([^18^F]**7**) [225] are ^18^F-labeled spirocyclic piperidine ligands derived from the lead compound, 1′-benzyl-3-methoxy-3*H*-spiro [2-benzofuran-1,4′-piperidine], with nanomolar affinity for σ_1_ receptors and excellent selectivity over σ_2_ and more than 60 other receptors, transporters, and ion channels [226,227,228]. Their binding affinity, lipophilicity (Log *D*), the free fraction in monkey plasma (*f*_P_) and the regional non-displaceable binding potential (*BP*_ND_) values are listed in Table 3. As expected, and consistent with results from rodent and pig studies [224,229,230,231,232], radioligands [^18^F]**4** and (*R*)-(+)-[^18^F]fluspidine, with subnanomolar affinity for the σ_1_ receptors, exhibited irreversible binding kinetics in monkey brain regions, with no appreciable washout during the 4 h scan. And thus, they were judged as unsuitable for human neuroimaging. On the other hand, [^18^F]**5** and (*S*)-(–)-[^18^F]fluspidine, with nanomolar affinity, displayed fast and reversible kinetics with good uptake and high specific binding signal in the monkey brain. These results again reinforce the principle in PET radioligand development: for a given target protein, higher affinity of the radioligand will lead to slower kinetics in the brain. Moreover, radioligand [^18^F]**5** displayed on average > 2 times higher *BP*_ND_ values than (*S*)-(–)-[^18^F]fluspidine [233]. Note that (*S*)-(−)-[^18^F]fluspidine has been used to evaluate σ_1_ receptor changes in patients with major depression [234] and the σ_1_ receptor occupancy by pridopidine in the human brain of healthy volunteers and in patients with Huntington’s disease [235]. However, there has been no report on the use of this radioligand to investigate the σ_1_ receptor in AD.

In the past decades, most of the σ_1_ receptor ligands have been designed and synthesized based on Glennon’s pharmacophore model (two hydrophobic regions and a basic nitrogen atom) [236]. Encouraged by the results from the spirocyclic piperidine radioligands described above, we undertook a study to develop a radioligand constructed from a novel scaffold and with optimal lipophilicity. Wuensch considered the benzene ring of the O-heterocycle of the spirocyclic piperidine derivative as the “primary hydrophobic region” and the phenyl group of the *N*-substituent as the “secondary hydrophobic region” of the σ_1_ ligands in Glennon’s pharmacophore model [237]. We replaced the spirocyclic piperidine moiety in [^18^F]**4** with a more hydrophilic group 1,4-dioxa-8-azaspiro [4.5]decane and simple piperidine [238]. The resulting ligands were found to maintain nanomolar affinity and subtype selectivity for σ_1_ receptors, indicating that removal of the benzene ring from the spiro(isobenzofuran piperidine) moiety still preserves the high affinity for the σ_1_ receptors [238]. Later, we replaced 1,4-dioxa-8-azaspiro [4.5]decane with 1,3-dioxane [239] or a tetrahydrofuran moiety [240], and found that these derivatives also maintained nanomolar affinity for the σ_1_ receptors. These findings demonstrated that smaller and less lipophilic moieties may serve as the “primary hydrophobic region” in the piperidine series of ligands. Thus the “primary hydrophobic region” in Glennon’s pharmacophore model appears to be more flexible, and can accommodate diverse structural moieties, not just those with an aromatic component.

Inspired by these new discoveries in Glennon’s pharmacophore model for σ_1_ receptor ligands, we designed and synthesized a novel radioligand [^18^F]FBFP with the smallest primary and secondary hydrophobic regions up to date for a σ_1_ receptor ligand. Gratifyingly, [^18^F]FBFP was found to have nanomolar affinity for the σ_1_ receptor, and high selectivity over the σ_2_ receptor, VAChT, and ten other receptors [240]. Studies in rodents and non-human primates indicated that this radioligand displayed fast, good brain uptake, favorable tissue kinetics, the highest plasma-free fraction and the highest specific binding signals in non-human primates among the σ_1_ receptor radioligands evaluated to date [240,241].

Similar to [^18^F]fluspidine, [^18^F]FBFP has a chiral center at the tetrahydrofuran moiety (denoted with an asterisk * in the structures shown in Figure 2), and thus is composed of two enantiomers. Enantiopure (*S*)-FBFP and (*R*)-FBFP were prepared from chiral synthesis with > 98% enantiomeric purity. In vitro evaluation demonstrated that (*R*)-FBFP with minus specific rotation behaved as an antagonist, while (*S*)-FBFP with plus specific rotation behaved as an agonist [242]. Both enantiomers possessed comparable low nanomolar affinity for the σ_1_ receptors and high selectivity over more than 40 other proteins. The enantiomerically pure radioligands (*S*)-(+)-[^18^F]FBFP and (*R*)-(−)-[^18^F]FBFP were obtained from their corresponding iodonium ylide precursors. Evaluation in rodents demonstrated excellent properties of both (*S*)-(+)-[^18^F]FBFP and (*R*)-(−)-[^18^F]FBFP with high brain uptake, high brain-to-blood ratios, high metabolic stability in the brain and high specific binding to the σ_1_ receptors [242]. In rhesus monkeys, both enantiomers display high brain uptake. Compared to (*S*)-(−)-[^18^F]fluspidine, both enantiomers exhibited much higher binding potential (*BP*_ND_) in rhesus monkeys (ranging from 9.6 (thalamus) to 27.7 (frontal cortex) for (*R*)-(−)-[^18^F]FBFP vs. 6.3 (cerebellum) to 14.8 (cingulate cortex) for (*S*)-(+)-[^18^F]FBFP) [243]. Although additional validation is required to assess utility in humans, both (*R*)-(−)-[^18^F]FBFP and (*S*)-(+)-[^18^F]FBFP, with the highest *BP*_ND_ values among the current available σ_1_ receptor ligands, meet all the requirements mentioned above for an optimal radioligand, and thus hold great potential for PET imaging and quantification of the σ_1_ receptor changes in AD patients. Both radioligands are currently undergoing evaluation in humans.

In summary, it appears that subnanomolar affinity for the σ_1_ receptors (*K*_i_ < 1 nM) will result in near-irreversible binding kinetics in the non-human primate brain. Radioligands [^18^F]**5**, [^18^F]**6**, [^18^F]**8** and [^18^F]**9** are suitable candidates for imaging σ_1_ receptors in humans.

**Table 3 ijms-24-12025-t003:** Binding affinity (*K*_i_, nM), Log *D*, *f*_P_ and *BP*_ND_ of the σ_1_ receptor radioligands.

Ligand	*K*_i_(σ_1_)	*K*_i_(σ_2_)	Selectivity	Log *D*	*f* _P_	*BP* _ND_
[^18^F]**4** ^a^	0.79	277	351	2.55	8%	-
[^18^F]**5** ^a^	2.30	327	142	2.50	18%	2.78–5.21
[^18^F]**6** ^a^	2.30	897	390	2.80	2%	0.77–1.85
[^18^F]**7** ^a^	0.57	1650	2895	2.80	2%	
[^18^F]**8** ^b^	2.26	299	127	0.76 ^c^	73%	6.3–14.8
[^18^F]**9** ^b^	1.61	246	152	0.76 ^c^	67%	9.6–27.7

^a^ From Ref. [233]. ^b^ From Ref. [243]. ^c^ From Ref. [240].

#### 3.2.3. Radioligands Targeting the σ_2_ Receptor

During the last decades, efforts in the development of σ_2_ receptor radioligands have been largely directed toward in vivo imaging of tumors in which upregulation of the σ_2_ receptor is found. Currently, [^18^F]ISO-1 ([^18^F]**10**) (Figure 3 and Table 4) is the only σ_2_ receptor radiotracer used in humans for tumor imaging [244,245]. However, it is not suitable for investigating neuronal σ_2_ receptors, due to its low brain uptake. There has been rekindled interest in the σ_2_ receptor as a therapeutic target for the treatment of neurologic and psychiatric diseases, especially AD. For example, the σ_2_ receptor antagonist CT1812 (Figure 1) is reported to prevent the binding of Aβ oligomers to neuronal receptors, and thus holds potential as a novel drug for the treatment of AD [193,194,246]. Hence, there remains an unmet clinical need to develop a suitable radioligand for neuroimaging of the σ_2_ receptor/TMEM97 in the human brain to investigate this target in AD progression, and to elucidate target engagement and the treatment mechanism of σ_2_ receptor-targeted drug candidates such as CT1812, in clinical trials.

Similar to what was found for σ_1_ receptor radioligands, the radiotracers for imaging σ_2_ receptors in the brain must meet the critical requirements outlined above. Due to undefined σ_2_ density in the brain of healthy human subjects and AD patients, the suitable affinity range required for effective imaging of σ_2_ receptors is yet to be defined. There has also been a paucity of ligands with high affinity and selectivity for the σ_2_ receptors. Although CT1812 is currently in Phase II clinical trial for treatment of mild-to-moderate AD, its affinity and subtype selectivity is only moderate [193]. Therefore, development of a suitable radioligand for neuroimaging of the σ_2_ receptors is even more challenging than the σ_1_ receptors. Nonetheless, there have been some recent activities in this endeavor, with several reports of brain-penetrant σ_2_ receptor radioligands, as depicted in Figure 3 [247,248,249,250].

In our search for highly selective radioligands for imaging σ_2_ receptors in the brain, we turned to the synthesis and evaluation of indole-based derivatives. Structure-activity relationship studies revealed that ligands with a four-carbon chain between the indole ring and the 6,7-dimethoxy-1,2,3,4-tetrahydroisoquinoline or 5,6-dimethoxyisoindoline pharmacophore displayed high *σ*_2_ receptor affinity and selectivity. Initial in vivo results indicated that radioligands with the 6,7-dimethoxy-1,2,3,4-tetrahydro-isoquinoline pharmacophore had lower brain uptake, brain-to-blood ratio, and *σ*_2_-specific binding [248,250]. Therefore, we focused on ligands with the 5,6-dimethoxyisoindoline pharmacophore to examine the influence of the fluoroethoxy group at different positions of the indole ring on tracer kinetics and specific binding. In biodistribution studies of three radioligands (named [^18^F]SYB4 ([^18^F]**11**), [^18^F]SYB5 ([^18^F]**12**), and [^18^F]SYB6 ([^18^F]**13**) (Table 4) in mice, they were found to readily enter the brain with good uptake (3.76–4.55% ID/g, 2 min) and high brain-to-blood ratios (10.6 for [^18^F]**11**, 30–60 min; 3.1 for [^18^F]**12** and 4.5 for [^18^F]**13**, 15 min), and to bind specifically to the *σ*_2_ receptor, indicating a significant achievement as the first set of radioligands demonstrated to be suitable for brain imaging purposes [248,250].

Ex vivo autoradiography and blocking studies demonstrated a high level of regionally heterogeneous specific binding of [^18^F]**11** in the mouse brain [248], with the distribution pattern clearly different from that we observed recently with a *σ*_1_ receptor radioligand [239]. Analysis results from a metabolism study in ICR mice indicated that the parent compound [^18^F]**11** or [^18^F]**13** was the predominant radioactive species (> 95%), indicating negligible entry of radiometabolites into the brain. Dynamic PET imaging and blocking studies in Sprague-Dawley rats confirmed regionally distinct distribution and high specific binding of radioligand [^18^F]**11** to the *σ*_2_ receptors in the rat brain [248].

**Table 4 ijms-24-12025-t004:** Binding affinity (*K*_i_, nM) and Log *D* of the σ_2_ receptor radioligands.

Ligand	*K*_i_(σ_1_)	*K*_i_(σ_2_)	Selectivity	Log *D*
[^18^F]SYB4 ^a^	371	1.79	207	2.43 ^b^
[^18^F]SYB5 ^a^	187	3.27	57	2.29 ^b^
[^18^F]SYB6 ^a^	376	2.63	143	2.17
[^18^F]ISO-1 ^c^	330	6.95	48	3.06
[^18^F]ISO-1 ^d^ [^18^F]ISO-1 ^e^	102 95.1	28.2 13.3	4 7	3.06 3.06

^a^ From Ref. [250]. ^b^ From Ref. [248]. ^c^ From Ref. [244]. ^d^ From Ref. [251]. ^e^ From Ref. [252].

In evaluation in monkeys, [^18^F]SYB4 ([^18^F]**11**) [253] and [^18^F]SYB6 ([^18^F]**13**) [254] exhibited fast and reversible kinetics, with peak SUV of 2.2–4.5 and 2.5–3.6, respectively, within 30 min in grey matter regions. The highest uptake was in the cerebellum and putamen, followed by similar uptake values in the hippocampus and caudate. Pretreatment with CM398 (0.2 mg/kg) reduced tracer uptake significantly across all brain regions. Regional *BP*_ND_ values ranged from 0.56 (amygdala) to 2.59 (cerebellum) for [^18^F]**11** and 0.92 (amygdala) to 2.11 (cerebellum) for [^18^F]**13**, indicating specific binding of both radioligands to the σ_2_ receptors [253,254]. These two radioligands represent the first generation of PET radiotracers demonstrated to be suitable for imaging and quantification of the σ_2_ receptor in the primate brain.

In addition to [^18^F]SYB4 and [^18^F]SYB6, several other putative σ_2_ receptor probes ([^18^F]**14**, [^11^C]**15**, [^11^C]**16**, and [^18^F]**17**, Figure 3) have been reported to have good brain uptake in mice. No further reports are available for their evaluation in non-human primates or humans.

## 4. Concluding Remarks

Hallmarks of human AD pathology (Aβ plaques and hyperphosphorylated tau protein tangles) have been proved to play critical and interrelated roles in AD pathogenesis and progression. However, increasing evidence has demonstrated the central role of the MAM dysfunctions in AD pathogenesis [255,256,257,258,259]. As a key chaperone situated at the MAM, the σ_1_ receptor is closely related to early Aβ generation, tau neurotoxicity, oxidative stress, and calcium dyshomeostasis [257]. Indeed, donepezil, the ‘gold standard’ acetylcholinesterase inhibitor (AChEI) in the symptomatic treatment of AD, has been found to have significant σ_1_ binding affinity (*IC*_50_ of 29.1 nM) [220], and to exert its anti-amnestic and neuroprotective activities against Aβ toxicity through activation of the σ_1_ receptors [139,220,260]. Rivastigmine, another AChEI used for AD treatment, is also found to derive its activity to enhance neuronal growth through interaction with the σ_1_ and σ_2_ receptors [178]. Finally, ANAVEX2-73, a σ_1_ receptor agonist, has been shown to provide significant and sustained improvement in cognitive function in mild-AD patients [189]. As a regulator of Aβ production and a surrogate biomarker for mitochondrial function, the σ_1_ receptor is increasingly viewed as playing a critical role in AD pathogenesis and progression, and thus holds great potential as an important target for therapeutic intervention and as a biomarker for early diagnosis, progression and monitoring of AD drug efficacy [49].

As a cholesterol-regulating gene, the σ_2_ receptor/TMEM97, PGRMC1 and LDLR form a trimeric complex (TMEM97/PGRMC1/LDLR) and behave as a binding site for monomeric and oligomeric amyloid β-peptide (1–42) (Aβ_1–42_) [44]. CT1812, a σ_2_ receptor antagonist in clinical trial for AD treatment, is reported to prevent the binding of Aβ oligomers to neuronal receptors [194,246], and to reduce the interaction between the σ_2_ receptor and Aβ oligomers in synapse in a dose-dependent manner [193,194,246]. The recently FDA-approved mAb for AD therapy, Leqembi, is shown to prevent the formation of Aβ oligomers which bind to the σ_2_ receptor/TMEM97-PGRMC1-LDLR complex [34]. Hence, the σ_2_ receptor/TMEM97 is considered a therapeutic target for AD.

For in vivo investigation of σ receptors, radioligands [^18^F]**5**, [^18^F]**6**, [^18^F]**8** and [^18^F]**9** have been proved to be suitable candidates for neuroimaging of σ_1_ receptors in non-human primates [233,241,243], with [^18^F]**6**, [^18^F]**8** and [^18^F]**9** in the clinical trials. Two radioligands, [^18^F]SYB4 and [^18^F]SYB6, have been found to be promising for neuroimaging of the σ_2_ receptors in rodents and non-human primates [248,250,253,254]. Their further investigation in clinical studies may finally afford us a radioligand suitable for imaging the σ_2_ receptor in humans. Advancement of these novel radioligands for imaging the σ_1_ and σ_2_ receptors in AD, especially in longitudinal studies, will visualize the changes in these receptors along the disease progression pathway, and thus help to elucidate the key roles of the σ_1_ and σ_2_ receptors in AD pathogenesis and progression, and to facilitate the development of effective therapeutic strategies for AD.

## Figures and Tables

**Figure 1 ijms-24-12025-f001:**
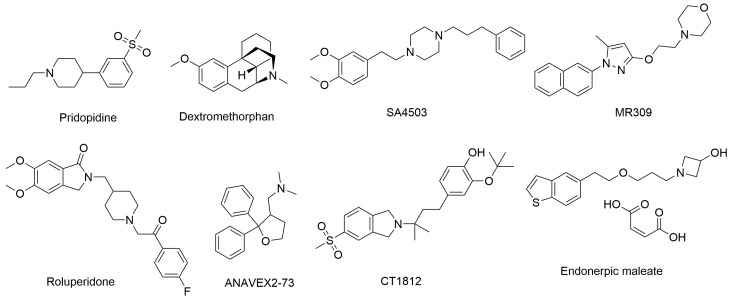
Chemical structures of representative σ_1_ and σ_2_ ligands with therapeutic potential in clinical trials.

**Figure 2 ijms-24-12025-f002:**
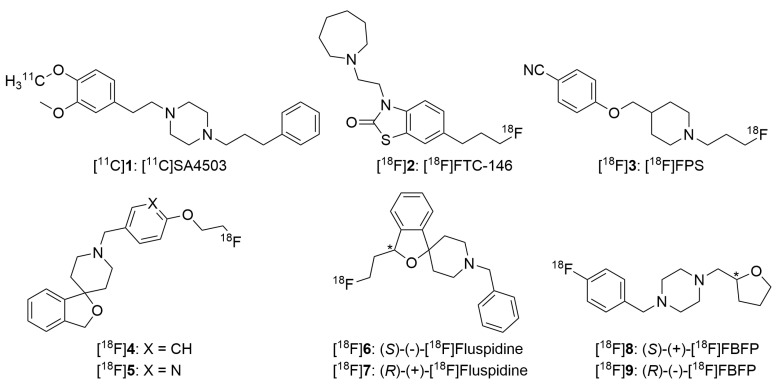
Chemical structures of σ_1_ receptor radioligands investigated in primates (* chiral center of the compound).

**Figure 3 ijms-24-12025-f003:**
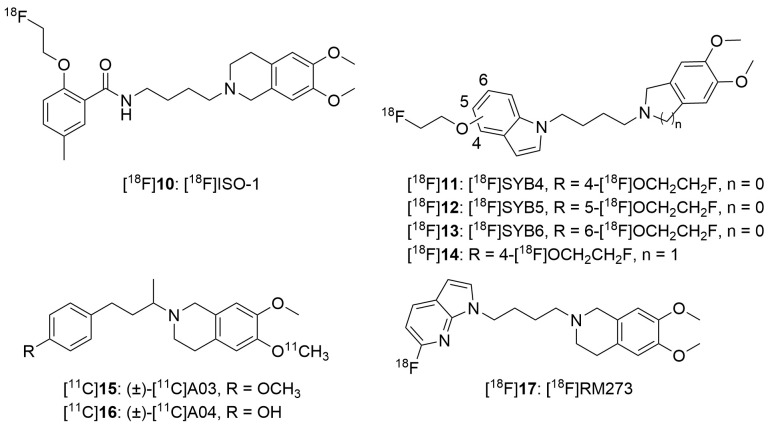
Chemical structures of potential radioligands for neuroimaging of σ_2_ receptors.

**Table 1 ijms-24-12025-t001:** Drugs available on the market for AD treatment.

Agent	Target ^a^	Mechanism ^b^	Chemical Structure
Donepezil	AChE	AChE reversible inhibition	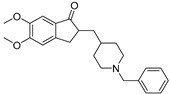
Galantamine	AChE	AChE reversible inhibition	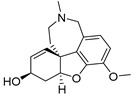
Rivastigmine	AChE	AChE reversible inhibition	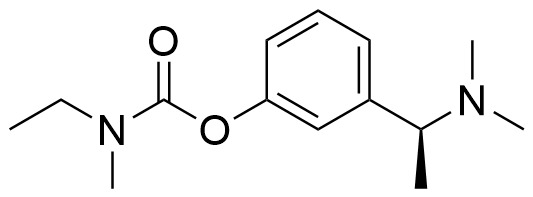
Memantine	NMDA receptors	NMDA non-competitive antagonist	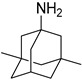
Aducanumab (Aduhelm) ^c^	Aβ plaque	mAb immunotherapy against Aβ	-
Lecanemab (Leqembi) ^d^	Protofibrillar and oligomeric forms of Aβ plaque	mAb immunotherapy against Aβ	-
GV-971 ^e^	Gut microbiota	yet to be fully elucidated	marine-derived oligosaccharide

^a^ AChE: acetylcholinesterase, NMDA: *N*-methyl-*D*-aspartate receptor, Aβ: β-amyloid. ^b^ mAb: monoclonal antibody. ^c^ Approved by the US Food and Drug Administration (FDA) for AD treatment on 7 June 2021 via the accelerated approval pathway. ^d^ Approved by FDA for AD treatment on 6 January 2023 via the accelerated approval pathway, and converted to traditional approval on 6 July 2023. ^e^ Approved by the National Medical Products Administration of China for AD treatment on 29 December 2019.

## Data Availability

Not applicable.

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
