# Peer review of "The Sigma Receptors in Alzheimer’s Disease: New Potential Targets for Diagnosis and Therapy"

_ijms, 2023, doi:10.3390/ijms241512025_

Round 1
Reviewer 1 Report
The reviewed publication presents sigma receptors as potential targets for the diagnosis and therapy of Alzheimer's disease. The work is meticulously prepared and provides detailed information about the receptors and their ligands, primarily focusing on their potential use in Alzheimer's disease.
Remarks:
It would be worthwhile to emphasize the epidemiological dimension of Alzheimer's disease in the introduction of the article. Additionally, based on the literature data, I suggest discussing the safety aspects of potential therapy (including adverse effects), considering the selectivity of ligands (receptor subtypes, type of pharmacological activity, and possibility of interaction with other receptors).
Minor editing of English language required.
Author Response
Response to Reviewer 1 Comments
Point 1: The reviewed publication presents sigma receptors as potential targets for the diagnosis and therapy of Alzheimer's disease. The work is meticulously prepared and provides detailed information about the receptors and their ligands, primarily focusing on their potential use in Alzheimer's disease.
Response 1: Thank you very much for your encouragement and kind recommendation.
Point 2: It would be worthwhile to emphasize the epidemiological dimension of Alzheimer's disease in the introduction of the article. Additionally, based on the literature data, I suggest discussing the safety aspects of potential therapy (including adverse effects), considering the selectivity of ligands (receptor subtypes, type of pharmacological activity, and possibility of interaction with other receptors).
Response 2: Thank you very much for the suggestion. The epidemiology has been reviewed in the literature (Eur J Neurol. 2018, 25(1) 59-70 and series of World Alzheimer Report). We only emphasize epidemiological dimension of Alzheimer's disease in the Introduction as follows.
Alzheimer’s disease is the most common form of dementia, accounting for 60-80% of all cases. According to the World Alzheimer Report 2022, about 55 million of people lived with dementia worldwide in 2019. This number is expected to rise to 139 million in 2050.
There are also many reviews discussing the safety aspects of potential therapy (Mol Med Rep. 2019 20(2) 1479-1487; Expert Opin Drug Saf. 2020 19(2) 147-157; Brain 2022 145(7) 2250-2275). Therefore, we only briefly describe the selectivity of ligands and discuss the safety of potential therapy in the introduction as follows (Here we have provided the detailed references in the manuscript).
Donepezil is a piperidine-based reversible inhibitor of acetylcholinesterase (AChE) inhibitor with high inhibitory activity (IC50(AChE) = 5.7 nM) and high selectivity over butyrylcholinesterase (BuChE) (IC50(BuChE) = 7138 nM, 1252-fold) (J Med Chem. 1995 38 4821−4829). Interestingly, donepezil binds to sigma-1 receptors in the living human brain at therapeutic doses (Int J Neuropsychopharmacol 2009 12(8) 1127-1131). Rivastigmine is a carbamate-based pseudo-irreversible (slowly reversible) dual inhibitor of both AChE and BuChE with low inhibitory potency (IC50(AChE) = 32100 nM, IC50(BuChE) = 390 nM, 82-fold) (Biomedicines 2022 10(7) 1510). However, this drug has no affinity for muscarinic, a- or b-adrenergic, or dopamine receptors or opioid binding sites (Prog Brain Res 1993 98 431–438). Galantamine is a reversible competitive inhibitor for AChE (IC50(AChE) = 350 nM) rather than BuChE (IC50(BuChE) = 18600 nM, 53-fold) (Life Sci 1990 46(21) 1553-1558) and allosteric modulator of nicotinic acetylcholine receptors (Eur J Pharmacol 2000 393 165-170). The most common adverse events of these cholinesterase inhibitors are gastrointestinal (GI) side effects. Other associated side effect is cardiovascular effects. However, they are generally well-tolerated and all are still considered first-line, symptomatic treatment of AD (for review, see ACS Chem Neurosci 2019 10 155−167; Mol Med Rep 2019 20(2) 1479-1487; Expert Opin Drug Saf 2020 19(2) 147-157; J Cent Nerv Syst Dis 2021 13 11795735211029113; Brain 2022 145(7) 2250-2275).
Memantine is a voltage-dependent, low affinity/fast off-rate and non-competitive N-methyl-D-aspartate receptor (NMDA) receptor antagonist (Curr Alzheimer Res 2005 2(2)155-65). It exerts its neuronal protective effects by inhibiting glutamate activity and is used for the treatment of moderate to severe AD alone or in combination with donepezil (Brain Behav 2020 10(11) e01831; JAMA 2019 322(16) 1589-1599; Pharmaceutics 2022 14(6)1117). Adverse effects of memantine have been found to be comparable to those with placebo, with the exception of an increased incidence of dizziness, headache, confusion, and constipation (Am J Geriatr Pharmacother 2004 2(4) 303-12).
Both Aducanumab (Aduhelm) and Lecanemab (Leqembi) are anti-amyloid monoclonal antibodies (mAbs) and approved under the accelerated approval pathway for treatment of Alzheimer patients with mild cognitive impairment. On July 6, 2023, FDA converted Lecanemab to traditional approval. And thus, Leqembi represents the first Ab-directed antibody to be converted from an accelerated approval to a traditional approval for the treatment of Alzheimer’s disease. Aducanumab is a human IgG1 monoclonal antibody preferably targeting Ab aggregates (Nature 2016 537(7618) 50-6). Lecanemab (Leqembi) is a humanized monoclonal IgG1 of the mouse mAb158 selectively binding to soluble Aβ protofibrils (N Engl J Med 2023 388(1) 9-21). Adverse effects with both medications include amyloid-related imaging abnormalities (ARIA) and infusion reactions (Nature 2016 537(7618) 50-6; J Prev Alzheimers Dis 2021 8(4)398-410; J Prev Alz Dis 2023 3(10) 362-377; N Engl J Med 2023 388(1) 9-21). Patients with apolipoprotein E ε4 (APOE4) gene carriers, especially APOE4 homozygotes, are at higher risk for ARIA such as brain swelling (edema or effusions) and bleeding (hemosiderin deposits) (J Prev Alz Dis 2023 3(10) 359-361; J Prev Alz Dis 2023 3(10) 362-377; Int J Mol Sci 2023 24(4) 3895).
Sodium Oligomannate Capsules (GV-971) was approved by the National Medical Products Administration (NMPA) of China in 2019 for treatment of mild-to-moderate AD (Drugs 2020, 80, 441-444). GV-971 was noted to have liver injury induction side effect (Authorea Preprints 2020). The other side effects of GV-971 has not been extensively reported in the literature. However, international multicenter clinical trials of GV-971 (Phase III) was stopped in May 2022.

Reviewer 2 Report
Overall, the review is well written and appropriately inclusive of current research on sigma receptors. There is a disconnect respective to “biomarker” capacity for AD. While there is a good deal of support for Sigma receptor as a therapeutic target in AD, biomarkers require a clear and consistent delineation of change over the course of the disease, which is not particularly well supported for either sigma-1 or -2. Nor is it clearly supported relative to “early” AD detection. Page-6 notes a 1.5-fold increase in sigma-2 in AD, which for biomarker purposes is a very narrow change, and just as critical is an increase which is counter to neurodegenerative processes. Moreover, the intracellular localization of the receptors is less than ideal for brain neuroimaging. Given the length of the ligand/radioligand section, a review simply focusing on sigma radioligands would be broad and still inclusive of AD as well as for other uses (e.g. tumors, etc.).
The Summary and Perspective section (4) have a good deal of information that could be placed more appropriately in other sections or is redundant and can be trimmed-out. A “review” is somewhat of a summary already, so re-summarizing various aspects is not necessary. Would suggest simply having a Conclusion section that incorporates the critical perspectives the authors wish to emphasize.
Minor
· Second paragraph of introduction has a single sentence on Aduhelm and Leqembi. A brief address as to controversy and side-effects (i.e. brain edema) may be worth noting.
· Table-1, Aducanumab’s FDA approval was also “fast-track” accelerated approval, similar to lecanemab.
· Are there any known endogenous ligands for the sigma receptors? A statement regarding the general lack of known endogenous ligands for sigma receptors
· Note of sigma-1 receptor being a “ligand-operated receptor chaperone”. Would be helpful to reading to explain this aspect.
· Grammar, last sentence on page-8 needs to be readjusted- “The optimal radioligands for imaging of the σ1 or σ2 receptors in the brain need to be fulfilled the following requirements..”
Author Response
Response to Reviewer 2 Comments
Point 1: Overall, the review is well written and appropriately inclusive of current research on sigma receptors. There is a disconnect respective to “biomarker” capacity for AD. While there is a good deal of support for Sigma receptor as a therapeutic target in AD, biomarkers require a clear and consistent delineation of change over the course of the disease, which is not particularly well supported for either sigma-1 or -2. Nor is it clearly supported relative to “early” AD detection. Page-6 notes a 1.5-fold increase in sigma-2 in AD, which for biomarker purposes is a very narrow change, and just as critical is an increase which is counter to neurodegenerative processes. Moreover, the intracellular localization of the receptors is less than ideal for brain neuroimaging. Given the length of the ligand/radioligand section, a review simply focusing on sigma radioligands would be broad and still inclusive of AD as well as for other uses (e.g. tumors, etc.).
Response 1: Thank you very much for your positive comments and kind suggestions. We agree with the reviewer that biomarkers require a clear and consistent delineation of change over the course of the disease. The sigma receptors are a relatively novel class of receptors that has yet to be completely understood. However, as a regulator of Ab production and surrogate biomarker for mitochondria function, the σ1 receptor is increasingly viewed as playing a critical role in AD pathogenesis and progression, and thus holds great potential as an important target for therapeutic intervention and a biomarker for early diagnosis, progression and monitoring of AD drug efficacy.
As a cholesterol-regulating gene, the σ2 receptor/TMEM97, PGRMC1 and LDLR form a trimeric complex (TMEM97/PGRMC1/LDLR) and behave as a binding site for monomeric and oligomeric amyloid b-peptide (1-42) (Ab1‒42) (PLoS One 2014 9 e111899). CT1812, a σ2 receptor antagonist in clinical trial for AD treatment, is reported to prevent the binding of Ab oligomers to neuronal receptors (Alzheimers Dement (N Y) 2019 5 20-26; Alzheimers Dement 2021 17 1365-1382), to induce synaptic degradation and reduce the interaction between σ2 receptor and Aβ in synapse by a dose-dependent manner (Int J Mol Sci 2023 24 6251; Alzheimers Dement (N Y) 2019 5 20-26; Alzheimers Dement 2021 17 1365-1382; ACS Med Chem Lett 2021 12 1389-1395). The recently FDA-approved mAb for AD therapy, Leqembi, is shown to prevent the formation of Aβ oligomers which is the binding site of σ2 receptor/TMEM97-PGRMC1-LDLR complex (N Engl J Med 2023 388 9-21). Hence the σ2 receptor/TMEM97 is considered a therapeutic target for AD.
Currently, there is a good deal of support for sigma receptor as a therapeutic target in AD. Optimal radioligands targeting the σ1 or σ2 receptors will probe the σ1 or σ2 receptor density changes in AD pathogenesis and progression and to monitor the effects of emerging therapeutics in clinical trials. So, in this review, we focus on the development of optimal PET radioligands for the s1 or s2 receptors, providing the opportunities for neuroimaging to elucidate the s1 and s2 receptors as novel biomarkers for early AD diagnosis, and monitoring of disease progression and AD drug efficacy.
Given that s1 receptor participates in various physiological and pathological processes, this receptor has been considered as a therapeutic target for a variety of diseases including amnesia and AD, Parkinson’s disease (PD), Huntington's Disease (HD), amyotrophic lateral sclerosis (ALS), retinal disease, multiple sclerosis (MS), major depressive disorder (MDD), stroke, pain and drug or alcohol addiction. In cancers, the s1 receptor is involved in tumor occurrence, development, metastasis and survival. Because there have been many reviews on the above diseases, we mainly discuss the application of sigma receptor radioligands in AD in this review.
Point 2: The Summary and Perspective section (4) have a good deal of information that could be placed more appropriately in other sections or is redundant and can be trimmed-out. A “review” is somewhat of a summary already, so re-summarizing various aspects is not necessary. Would suggest simply having a Conclusion section that incorporates the critical perspectives the authors wish to emphasize.
Response 2: We have followed your suggestions and deleted the Summary and Perspectives section (4) and some information in this part have been placed in “Development of radioligands for neuroimaging of σ Receptors” section highlighted with yellow color in the revised version. We emphasize the critical perspectives in Concluding remarks.
Point 3: Second paragraph of introduction has a single sentence on Aduhelm and Leqembi. A brief address as to controversy and side-effects (i.e. brain edema) may be worth noting.
Response 3: A brief address as to controversy and side-effects (i.e. brain edema) of Aduhelm and Leqembi has been added in the revised version as follows.
Both Aducanumab (Aduhelm) and Lecanemab (Leqembi) are anti-amyloid monoclonal antibodies (mAbs) and approved under the accelerated approval pathway for treatment of Alzheimer patients with mild cognitive impairment. On July 6, 2023, FDA converted Lecanemab to traditional approval. And thus, Leqembi represents the first Ab-directed antibody to be converted from an accelerated approval to a traditional approval for the treatment of Alzheimer’s disease. Aducanumab is a human IgG1 monoclonal antibody preferably targeting Ab aggregates (Nature 2016 537(7618) 50-6). Lecanemab (Leqembi) is a humanized monoclonal IgG1 of the mouse mAb158 selectively binding to soluble Aβ protofibrils (N Engl J Med 2023 388(1) 9-21). Adverse effects with both medications include amyloid-related imaging abnormalities (ARIA) and infusion reactions (Nature 2016 537(7618) 50-6; J Prev Alzheimers Dis 2021 8(4) 398-410; J Prev Alz Dis 2023 3(10) 362-377; N Engl J Med 2023 388(1) 9-21). Patients with apolipoprotein E ε4 (APOE4) gene carriers, especially APOE4 homozygotes, are at higher risk for ARIA such as brain swelling (edema or effusions) and bleeding (hemosiderin deposits) (J Prev Alz Dis 2023 3(10) 359-361; J Prev Alz Dis 2023 3(10) 362-377; Int J Mol Sci 2023 24(4) 3895).
Point 4: Table-1, Aducanumab’s FDA approval was also “fast-track” accelerated approval, similar to lecanemab.
Response 4: We have added the “fast-track” accelerated approval of Aducanumab’s FDA approval in Table 1.
Point 5: Are there any known endogenous ligands for the sigma receptors? A statement regarding the general lack of known endogenous ligands for sigma receptors.
Response 5: We have added a statement mentioned above regarding the general lack of known endogenous ligands for sigma receptors in 2.1 The sigma (s) receptors as follows.
Currently, there is no consensus on endogenous ligands for the s1 receptor (for review, see Trends Pharmacol Sci. 2019 Sep, 40(9), 636-654.), even though some candidates such as the hallucinogen N, N-Dimethyltryptamine (Science 2009 323(5916) 934-937) have been proposed. Up to date, two putative endogenous ligands: histatin-1 (FEBS J. 2021, 288(23), 6815–6827) and 20(S)-hydroxycholesterol (20(S)-OHC) (Nat. Chem. Biol. 2021, 17, 1271–1280) for the s2 receptor have been reported.
Point 6: Note of sigma-1 receptor being a “ligand-operated receptor chaperone”. Would be helpful to reading to explain this aspect.
Response 6: Sigma-1 receptor has approved to be “ligand-operated receptor chaperone”. We have added a sentence to briefly explain this aspect in the manuscript as follows.
The s1 receptor has proved to be a unique “ligand-operated receptor chaperone” that is regulated by the agonist/antagonist activity of endogenous or synthetic ligands.
Point 7: Grammar, last sentence on page-8 needs to be readjusted- “The optimal radioligands for imaging of the σ1 or σ2 receptors in the brain need to be fulfilled the following requirements.”
Response 7: Many thanks for your suggestion. This is changed as follows.
The optimal radioligands for imaging of the σ1 or σ2 receptors in the brain need to meet the following requirements.
